# Diagnostic Performance of F18-FDG PET/CT in Male Breast Cancers Patients

**DOI:** 10.3390/diagnostics11010119

**Published:** 2021-01-13

**Authors:** Andra Piciu, Doina Piciu, Narcis Polocoser, Anita A. Kovendi, Iulia Almasan, Alexandru Mester, Dragos-Stefan Morariu, Calin Cainap, Simona Sorana Cainap

**Affiliations:** 1Department of Medical Oncology, Iuliu Hatieganu University of Medicine and Pharmacy, 400012 Cluj-Napoca, Romania; piciuandra@gmail.com (A.P.); narcispol@gmail.com (N.P.); kovendianita@yahoo.com (A.A.K.); calincainap2015@gmail.com (C.C.); 2PhD School of Iuliu Hatieganu, University of Medicine and Pharmacy, 400012 Cluj-Napoca, Romania; almasan.iulia@gmail.com; 3Department of Endocrine Tumors and Nuclear Medicine, Institute of Oncology, 400012 Cluj-Napoca, Romania; 4Departement of Oral Health, University of Medicine and Pharmacy, 400012 Cluj-Napoca, Romania; alexandrumester@yahoo.com; 5Department of Surgery, University of Medicine and Pharmacy, 400012 Cluj-Napoca, Romania; dragosstefanmorariu@gmail.com; 6Department of Mother and Child, Iuliu Hatieganu University of Medicine and Pharmacy, 400012 Cluj-Napoca, Romania; simona.cainap@yahoo.com

**Keywords:** male breast cancer, synchronous malignancies, F18-FDG PET/CT

## Abstract

Introduction: F18-FDG PET/CT is the most important hybrid imaging used in the diagnostic, staging, follow-up, and treatment evaluation response in cancer patients. However, it is well-known that in breast cancer the use of F18-FDG is not included in the first line protocol of initial diagnostic, both in female and male breast cancer patients. F18-FDG PET/CT is a valuable tool to provide information on extra-axillary lymph node involvement, distant metastases, and other occult primary cancers. This study assesses F18-FDG PET/CT systemic staging in male patients with diagnosed breast cancer and determines detection rates for unsuspected distant metastases and synchronous malignancies. Methods: We analyzed a number of 170 male patients with breast cancer, seen between 2000–2020, in a tertiary center. From this group, between 2013–2020 a number of 23 patients underwent F18-FDG PET/CT. Rates of upstaging were determined for each case and the detection of other primary malignancies was analyzed. Results: Median age of male breast cancer group was 61.3 y (range, 34–85 y), most had intraductal carcinoma (82.4%) and unsuspected distant metastases, which increased patient stage to IV, observed in 27%. In 4 out 23 patients (17.4%), F18-FDG PET/CT identified synchronous cancers (2 prostate cancers, 1 thyroid and 1 colon cancer). Conclusion: F18-FDG PET/CT is a valuable tool to provide information on extra-axillary lymph node involvement, distant metastases, and other occult primary cancers. Baseline F18-FDG PET/CT has a substantial impact on the initial staging and on clinical management in male breast patients and should be considered for use in newly diagnosed patients.

## 1. Introduction

The PET/CT hybrid imaging is an essential tool in primary diagnostic, staging, follow-up, and treatment response evaluation of many malignant diseases. Compared to other cancers, in breast cancer the use of F18-FDG PET/CT has proven clinical value in selected cases: in patients with newly diagnosed breast cancer, F18-FDG PET/CT is recommended for stage III disease [1] and for restaging, follow-up, and treatment response evaluation. Some studies recommend due to the detection of unsuspected distant metastases, fact that alters treatment and prognosis, that the guidelines should consider adding in patients with stage IIB breast cancer the systemic staging with F18-FDG-PET/CT at the time of the initial diagnosis [2,3,4,5]. F18-FDG PET/CT reduced false-positive risk by half and decreased workup of incidental findings, allowing earlier treatment and being cost-effective [4]. Approximately 37% of patients with clinical stage IIA-IIIC breast cancer who underwent F18-FDG PET/CT before preoperative systemic therapy showed more extensive disease, including 23% with more extensive nodal metastasis and 14% with distant metastasis [5]. The male breast cancer represents only 1% of all breast cancers [3,6] and the literature is by far less consistent than in women, studies have limited number of subjects and many are case reports [6,7,8,9,10,11]. In a recent study published by Ulaner et al. [7] it is underlined that the role of F18-FDG PET/CT is based on female guidelines and that an analysis of the value of F18-FDG PET/CT for systemic staging of male breast cancer is warranted. In this study, we retrospectively search the database of a tertiary cancer center to evaluate the value of F18-FDG PET/CT in male breast cancer patients, in detecting the presence of synchronous tumors.

## 2. Materials and Methods

### 2.1. Study Design

This retrospective single-institution study was performed on a database of a Romanian tertiary cancer center, the Institute of Oncology “Prof. Dr. Ion Chiricuță” Cluj-Napoca (IOCN) searching for patients with male breast cancer between January 2000 and November 2020. The inclusion criteria were: male breast cancer histological confirmed; hormonal receptors status: estrogen receptor (ER), progesterone receptor (PR), and human epidermal growth factor receptor 2 (HER2) described; we found an important number of patients (43%) without HER2 status defined, because of the lack of determinations in the first decade of the database; all patients signed the institutional informed consent both for diagnostic and treatment procedure, and for the use of their data in scientific reports, with personal data protection respected. The study protocol was approved by the Ethics Committee of the Institute of Oncology “Prof.Dr.I.Chiricuta” Cluj-Napoca, approval number 174/28.02.2020. The study was conducted according to the principles of the Declaration of Helsinki, the International Conference on Harmonization Guideline on Good Clinical Practice, the Romanian laws and regulations.

Epidemiologic data on this cohort and the survival parameters are reported in this study. Between August 2013–November 2020 from this cohort 23 patients have been evaluated by F18-FDG PET/CT in different moments of the disease: staging, restaging, and during monitoring. In this group the hybrid imaging parameters were analyzed: presence or absence of pathologic F18-FDG uptake, number of lesions, standardized uptake values (SUV). All lesions suspected for synchronous primary tumors were registered and followed for their histological confirmation.

### 2.2. PET/CT Imaging

All hybrid images were obtained with a GE Optima 560 PET/CT, GE Healthcare USA; F18-FDG PET/CT studies were performed after a period of fasting of 6 h; the blood glucose levels between 76 mg/dL–148 mg/dL were measured before the F18-FDG injection; injected doses had average activity of 241.4 MBq (Min 148–Max 380 MBq). F18-FDG activities were calculated according to the patients’ weight and by the European Association of Nuclear Medicine (EANM) procedure guidelines for tumor imaging, version 2.0 [12] and the examination was performed after an uptake period of 50–70 min. CT images, with a slice thickness of 3.75 mm, were acquired using a low-dose protocol (100–130 kV, 50–100 auto mA, index noise of 20%) in order to reduce the irradiation dose for patients. F18-FDG PET/CT images were evaluated by a nuclear medicine physician and a radiologist. For all F18-FDG PET/CT studies, SUVlbm (the standardized uptake value lean body mass) a semi quantitative parameter for F18-F18-FDG uptake calculation respecting a standard protocol on the work station (Volumetrix for PET/CT) was used.

### 2.3. Statistical Analysis

Statistical analysis was performed using GraphPad Prism 6.0 software. We calculated means and standard deviations.

## 3. Results

A total of 170 male patients diagnosed with breast cancer from 2000–2020 were analyzed. In a period of 20 years, the male breast cancer represents 0.1% from all breast cancer reported in our institution. The mean ± SD age was 61.34 ± 11.19 year-old. The majority, 140 patients (82.4%) had the histology of intraductal carcinoma (IDC), 7 cases (5%) had invasive lobular carcinoma (ILC), 4 cases (2.4%) had in situ ductal carcinoma, and 19 cases (11.2) had other forms. Regarding the receptor immunohistochemistry, 10 patients (5.6%) were ER negative, 138 cases (81.2%) were ER positive, and in 22 patients (12.9%) the ER status was not available. Twelve patients (7%) were PR negative and 137 (80.6%) were PR positive, in 21 patients (12.3%) PR status was not available. HER2 was not available in 79 cases (46.5%); was 0 in 43 cases (25.3%); 1 in 26 cases (15.3%); 2 in 15 cases (8.8%); 3 in 7 cases (4.1%). None of our patients was triple negative. At the moment of the report 95 patients were alive (55.8%), 65 patients (38.2%) were dead, and in 10 cases the survival data were not available. Thirty-two patients (60.3%) in stage III of the disease were alive at the moment of the study, treated by radical mastectomy and ALND, radiotherapy and chemotherapy; 8 patients with stage II and III presented metastatic recurrence of the disease during the first 2 years, but are alive. Among the 65 dead patients, 21 were in stage III, 5 patients were diagnosed in stage IV, and 39 of them had an unknown cause of the death, less likely related to their disease.

The main characteristics of the database is presented in Table 1.

The analysis of this report revealed an unusual high number of patients with skin and subcutaneous metastases, 6 out of 17 (35.3%) in the group of subjects with distant metastases, the majority being discovered on F18-FDG PET/CT scans, Figure 1.

From the above mentioned database, between 2013 and 2020, a number of 23 patients were evaluated by hybrid imaging F18-FDG PET/CT, Figure 2. The male breast cancers represented 2.06% from all scans performed for breast carcinomas. The percentage of scans is considerably higher (2.06%) compared to the percent of male breast cancer among all breast carcinoma cases (0.1%). A number of 4 patients (17.4%) were discovered with multiple primary malignancies during the metabolic scan, confirmed in histology: 1—thyroid cancer, 1—prostate cancer, 1—Hodgkin lymphoma, and 1—colon cancer.

The characteristics of patients followed by F18-FDG PET/CT scan are presented in Table 2.

The cohort of patients submitted for F18-FDG PET/CT scan has the mean age 60.69 ± 9.216 year-old, age slightly lower compared to the whole cohort of male patients with breast cancers.

F18-FDG PET/CT scans was performed for restaging in 5 cases (21.7%), for the increase of tumor markers in 13 (56.5%), assessment for response to therapy in 2 (8.7%), and during follow-up in 3 cases (13%). The scans demonstrated new, unknown lesions suggestive for metastases in 12 (52.2%) out of 23 patients. The unsuspected metastatic sites included: skeleton, liver, sub-pectoral and mediastinal lymphnodes, lungs, muscles, subcutaneous, and skin. The rate of upstaging was 26%, occurring in 6 patients.

The synchronous primary malignancies were demonstrated in 4 cases, confirmed in histology after: total thyroidectomy for papillary thyroid carcinoma; cervical lymphnode biopsy for Hodgkin lymphoma; prostatic biopsy for prostate cancer and left hemicolectomy for colon adenocarcinoma. In Figure 3, Figure 4, Figure 5 and Figure 6 we present the images of patients with synchronous tumors detected in F18-FDG PET/CT scans during restaging, tumor marker increased, or follow-up.

## 4. Discussion

In a period of 20 years, 2000–2020, we report a number of 170 cases of male breast cancer patients, representing 0.1% from all breast cancers reported in our institution. It is surprising the very low percent, compared with other published data, which report an incidence among all other breast cancers 10-fold higher, respectively 1% [3,6]. The mean age was 61.34 ± 11.198, slightly lower than the age reported in other studies [6,13] and the majority of cases were in a more advanced stage (64.8% in stage III and IV) compared with other studies [1,6,7,13,14], suggesting the lack of awareness about this cancer, in male population. Similarly to other published studies, the majority of cases 82.4% had the histology of intraductal carcinoma and more than 80% of the cases were estrogen and progesterone receptors (ER/PR) positive. None of our cases expressed triple negative histology. At the moment of this retrospective study 55.8% patients were alive. Despite the low incidence of this pathology in male, the aggressiveness and severity of this malignancy needs special attention. The F18-FDG PET/CT hybrid imaging is an essential tool in primary diagnostic, staging, follow-up, and treatment response evaluation of malignant diseases, having an increasing role in the management of breast cancers. Compared with other studies, in our male breast cancer cohort, this method was not used for staging; the scanning was performed for restaging in 21.7% and the majority of scans were indicated because of the increase of the tumor markers (56.5%). Just a few cases were referred for treatment response assessment (8.7%) and similarly for follow-up, underling the lack of evidences, both in male and female breast cancers, for the utility of the method in the long-term monitoring of the disease. Current guidelines do not recommend intensive surveillance, including F18-FDG PET/CT, in asymptomatic breast cancer patients [15]. Our retrospective review of male patients with breast cancer demonstrates that F18-FDG PET/CT detects unsuspected distant metastases at a very high rate of 52.2%, which is a rate above those published by Ulaner et al. [7] and other authors [6,9]; this fact might be explained by the advanced stages of our cases and also due to the complete lack of initial, baseline F18-FDG PET/CT imaging despite the advanced diseases. The rate of upstaging was lower than in other studies, occurring in 6 patients, mainly because our group consisted of patients in advanced stages; this fact suggested the role of F18-FDG PET/CT in the initial staging of male patients with breast cancer. The treatment plan for these patients needed to be changed from surgical management to systemic therapy without radical surgical procedure in 7 patients. The metastatic sites were confirmed through biopsy in 5 out of 12 patients with unsuspected metastases, in the rest of patients the tumor markers in dynamic evolution and the response to therapy were arguments for the positive diagnostic.

The number of 4 patients detected with synchronous tumors in F18-FDG PET/CT was a considerable high number. In this study we reported the occurrence of thyroid cancer and prostate cancer in association with breast cancer. In female, the association of thyroid–breast cancers as multiple primary malignancies is well studied, without having a clear scientific accepted explanation, except the more systematic screening. The interactions between thyroid and breast disorders are based on hormonal and cellular receptor mechanisms [16,17]. Thyroid cancer survivors also have been found to develop breast cancer early, have more estrogen and progesterone receptor positive tumors, and have a greater incidence of mixed invasive cancer [18]. Estrogen receptors have been found in thyroid tissue [19]. The histology of the breast cancer that develops after thyroid cancer is different than the general population, with a greater percentage of ER/PR-positive tumors [19]. Although indicated by many studies, an association between breast and thyroid cancer still remains controversial [20]. The high percentage of male patients with ER/PR positive might lead to the conclusion that male breast cancer patients need a careful examination and evaluation of the thyroid, in order to exclude the thyroid malignancy.

Regarding the association with prostate cancer a review of the existing literature shows that both prostate and breast cancers are typically hormone-dependent tumors and have similarities regarding etiology, epidemiology, and treatment approaches [21]. Alteration of estrogen to testosterone ratio is another possible explanation for the increased risk of hormone-sensitive cancers [22]. In this study we found breast cancer associated both with thyroid and prostate cancer. Despite the limited number of patients, we need to analyze the option of screening for thyroid and prostate cancers in male breast cancer patients.

The limitations of the study consist in the low number of patients in this cohort, but male breast cancer is an uncommon malignancy, and also, the number of synchronous malignancies detected in F18-FDG PET/CT was very limited; considering these facts, every study will contribute to the existing databases. The retrospective single-institution study is another limitation. Another limit of the study was the lack of baseline examinations, but F18-FDG PET/CT scans succeeded to correctly modify the staging and to determine the correct management of patients with multiple primary cancers.

## 5. Conclusions

F18-FDG PET/CT is a valuable tool to provide information on extra-axillary lymph node involvement, distant metastases, and other occult primary cancers in the evaluation of male breast cancer. Baseline F18-FDG PET/CT has a substantial impact on initial staging and on clinical management in male breast patients and should be considered for use in newly diagnosed patients. Despite the rarity of the pathology, the occurrence of multiple primary cancers needs to be carefully evaluated, in this light, the role of hybrid imaging being essential.

## Figures and Tables

**Figure 1 diagnostics-11-00119-f001:**
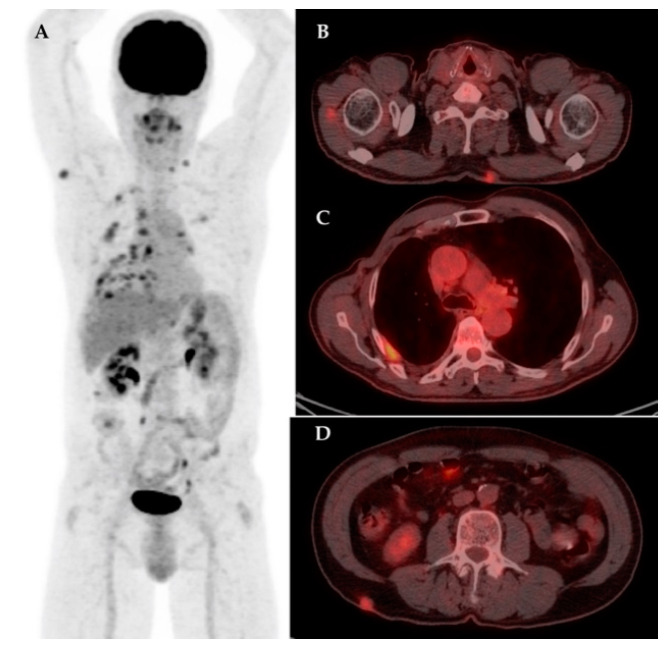
Male patient with right breast cancer, with right radical mastectomy and axillary lymphnode dissection, with multiple sites of increased F18-FDG uptake confirmed as breast carcinoma metastases. MIP image on F18-FDG PET/CT (**A**); axial section at cervical level showing muscle and posterior cervical subcutaneous lesions with increased F18-FDG uptake confirmed metastases on biopsy (**B**); axial section at thorax level showing pathologic increased F18-FDG uptake in the right pleura (**C**); axial section at lumbar level showing right posterior lumbar subcutaneous lesion with increased F18-FDG uptake confirmed metastases on biopsy (**D**).

**Figure 2 diagnostics-11-00119-f002:**
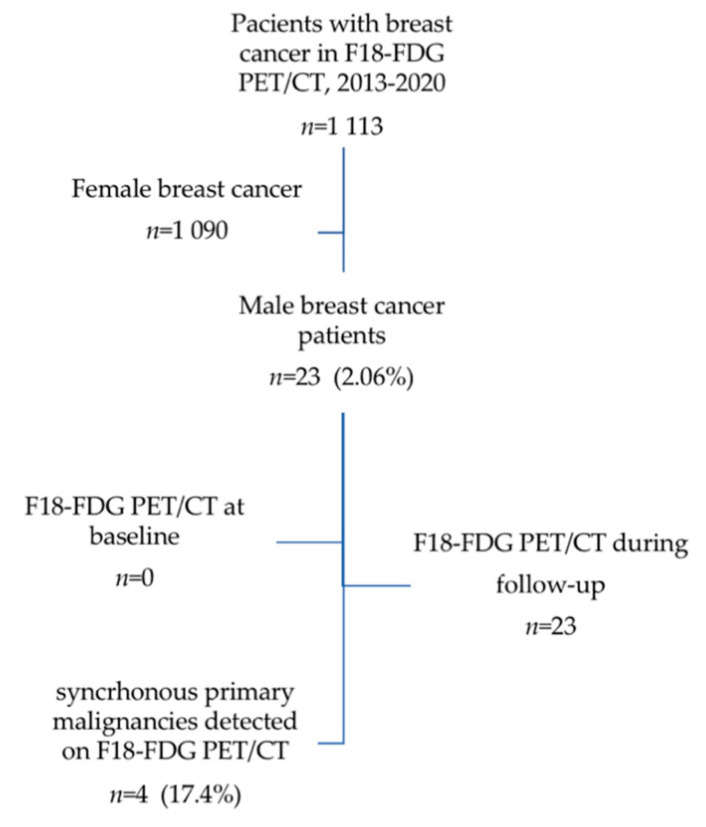
Male breast cancer patients evaluated by F18-FDG PET/CT 2013–2020.

**Figure 3 diagnostics-11-00119-f003:**
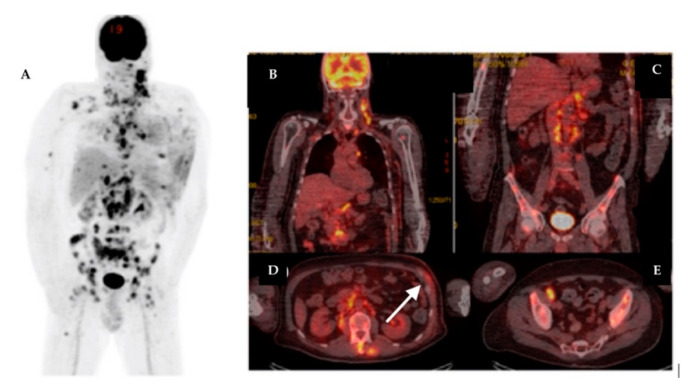
Male patient with left breast cancer and multiple metastatic lesions. Synchronous confirmed Hodgkin lymphoma. MIP image on F18-FDG PET/CT (**A**); coronal section at thorax level showing multiple cervical and mediastinal pathologic lymph nodes, with increased F18-FDG uptake (**B**); coronal section at abdominal level showing multiple pathologic lymph nodes with increased F18-FDG uptake in retroperitoneal area and multiple bone lesions, confirmed as metastatic from breast cancer (**C**); axial section at upper abdominal level showing multiple pathologic lymph nodes with increased F18-FDG uptake in retroperitoneal area and multiple bone and muscle lesions (**D**) and infiltration of left breast area posttherapy, with recurrence of the tumor (white arrow); axial section at pelvic level showing pathologic lymph node with increased F18-FDG uptake in right external iliac area and multiple bone lesions (**E**).

**Figure 4 diagnostics-11-00119-f004:**
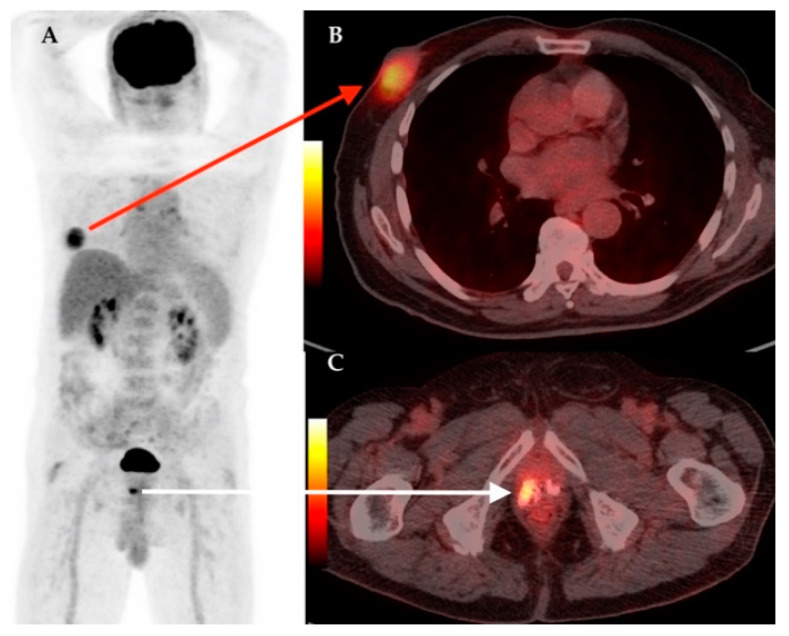
Male patient with right breast cancer. Synchronous confirmed prostate adenocarcinoma. MIP image on F18-FDG PET/CT (**A**); axial section at thorax level showing right breast lesion with increased F18-FDG uptake confirmed breast carcinoma on biopsy (**B**, red arrow); axial section at pelvic level showing pathologic increased F18-FDG uptake in right prostate lobe area with multiple calcification (**C**, white arrow), confirmed prostate adenocarcinoma.

**Figure 5 diagnostics-11-00119-f005:**
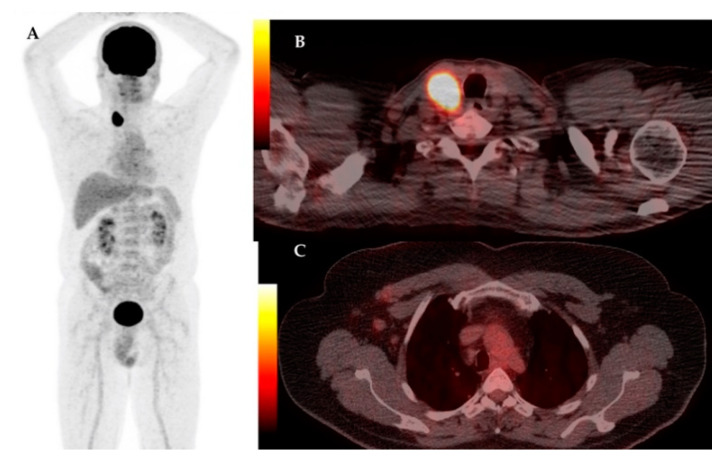
Male patient with right breast cancer, with right axillary lymphnodes suggestive for metastases. Synchronous confirmed papillary thyroid carcinoma of the right lobe. MIP image on F18-FDG PET/CT (**A**); axial section at cervical level showing thyroid right lobe lesion with increased F18-FDG uptake confirmed thyroid carcinoma on total thyroidectomy (**B**); axial section at thorax level showing pathologic slightly increased F18-FDG uptake in right axilla in the lymphnodes, confirmed metastatic **(C**).

**Figure 6 diagnostics-11-00119-f006:**
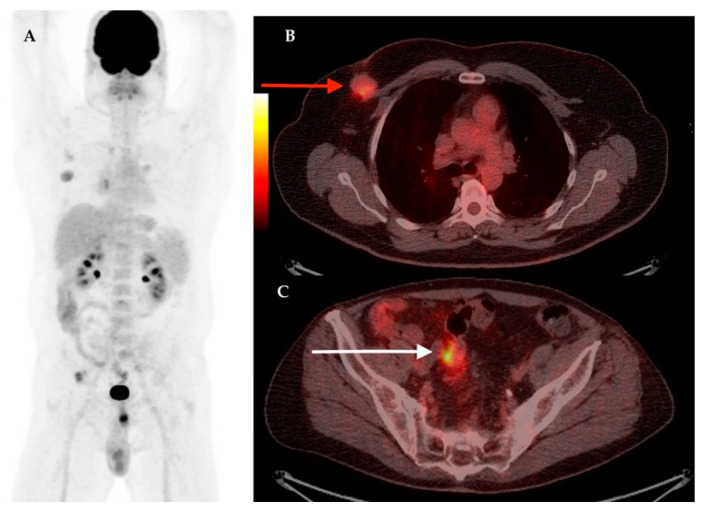
Male patient with right breast cancer. Synchronous confirmed colonic adenocarcinoma. MIP image on F18-FDG PET/CT (**A**); axial section at thorax level showing right breast lesion with increased F18-FDG uptake confirmed breast carcinoma on biopsy (**B**, red arrow); axial section at pelvic level showing pathologic increased F18-FDG uptake in the colon (**C**, white arrow), confirmed colon adenocarcinoma in biopsy at colonoscopy.

**Table 1 diagnostics-11-00119-t001:** Characteristics of male patients with breast cancer in IOCN 2000–2020.

Age (Years)	61.34
**Tumor size**	
T in situ	4
T1	20
T2	43
T3	16
T4	68
NA	19
**Grading (G)**	
3 (High)	35
2 (Intermediate)	92
1 (Low)	18
NA	35
**Lymph nodes (N) status**	
**N0**	13
**N1**	48
**N2**	33
**N3**	15
**NA**	61
**Metastasis (M)**	
**M0**	151
**M1 (skin, bone, lung, brain, hepatic)**	6/5/4/1/1
**NA**	2
**Receptors status**	
ER+/ER−/NA	138/10/22
PR+/PR−/NA	137/12/21
HER2–0/1/2/3/NA	43/26/15/79
**Stage**	
**IA,B**	6
**IIA**	17
**IIB**	9
**IIIA**	12
**IIIB**	27
**IIIC**	14
**IV**	6
**NA**	79
**Surgical Therapy**	
Radical mastectomy	149
Breast conservative surgery	2
No surgery	19
ALND	19
SLND	2
No axillary evaluation	19
**External Beam Radiotherapy**	
Yes	74
No	96
**Neoadjuvant chemotherapy**	
Yes/No	55/115
**Adjuvant chemotherapy**	
Yes/No	62/118
**Adjuvant antiestrogenic therapy**	
Yes/No	102/68
**Alive/Death/NA**	95/70/5

NA-not available data.

**Table 2 diagnostics-11-00119-t002:** The characteristics of patients followed by F18-FDG PET/CT scan.

No.	Age	Histo.	Surgery	CHT	EBR	Stage	Mets *	New Mets **	SUV1 Max	Synchronous Cancer	SUV2 Max
1	66	IDC	RM + SLND	Yes	Yes	IV	LN, Bn	LN, Lu	7.04	–	–
2	60	IDC	RM + ALND	NA	Yes	IIIA	–	–	–	–	–
3	66	IDC	RM + ALND	–	–	IIA	–	–	–	–	–
4	61	IDC	RM	–	–	IIA	–	–	–	–	–
5	70	IDC	RM + ALND	–	–	IB	–	Lu, Su	3.37	Colon cancer	5.27
6	52	IDC	RM + ALND	–	–	IIA	–	LN	4.77	–	–
7	47	IDC	Biopsy	–	Yes	IIIC	LN	Bn, Sk	18.36	–	–
8	52	IDC	RM + ALND	Yes	Yes	IV	LN, Bn	Lu	4.38	–	–
9	67	IDC	RM + ALND	Yes	Yes	IIA	–	–	–	–	–
10	57	IDC	RM + ALND	NA	NA	IB	–	LN	1.55	–	–
11	47	IDC	RM + ALND	–	–	IIB	–	LN, Bn	10.61	–	–
12	79	PBC	RM + ALND	Yes	Yes	IB	–	–	–	–	–
13	67	IDC	RM + ALND	–	–	IIB	LN	LN, L	4.43	–	–
14	68	IDC	RM + ALND	Yes	Yes	IIIB	LN	LN, Bn	7.04	Prostate cancer	5.01–
15	52	IDC	RM + ALND	Yes	Yes	IIB	–	–	–	–	–
16	64	ILC	RM + ALND	–	–	IIA	–	–	–	–	–
17	61	IDC	RM + ALND	Yes	Yes	IIIB	LN	LN, Lu, M, Bn	3.1	Hodgkin Lymphoma	11.6
18	42	IDC	RM + ALND	Yes	Yes	IIIC	–	–	–	–	–
19	74	IDC	RM + ALND	Yes	Yes	IIIC	LN, Bn	LN, Sk, M	6.21	–	–
20	59	IDC	RM + ALND	Yes	Yes	IIIB	–	LN, Sk	8.6	–	–
21	61	IDC	RM + ALND	Yes	Yes	IIC	LN	–	2.38	Thyroid cancer	23.1
22	69	IDC	RM + ALND	NA	NA	IIA	–	–	–	–	–
23	55	ILC	RM + ALND	Yes	Yes	IIB	–	–	–	–	–

Histo–histology; IDC-ductal invasive carcinoma; ILC–invasive lobular carcinoma; PBC–papillary breast carcinoma; RM–radical mastectomy; ALND–axillary lymphnode dissection; SNL–sentinel lymphnode biopsy; CHT–chemotherapy; EBR–external beam therapy; Mets *–known metastases; New mets **–unsuspected metastases detected on F18-FDG PET/CT; LN–lymphnode; L–liver; Lu–lung; Sk–skin; Su–subcutaneous; Bn–bone; B–brain; M–muscle; SUV1Max–maximum standardized uptake value in the hottest known metastasis; SUV2Max–maximum standardized uptake value in the lesion suspected as synchronous primary cancer; NA–unavailable data.

## Data Availability

The data presented in this study are available on request from the corresponding author.

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
