# Peer review of "Diagnostic Performance of F18-FDG PET/CT in Male Breast Cancers Patients"

_diagnostics, 2021, doi:10.3390/diagnostics11010119_

Round 1
Reviewer 1 Report
The article is easy to read, and images are very didactical, however English needs corrections.
The main drawback of the paper is the title, that is misleading, since the interest of the paper is not only directed to synchronous tumours but covers the role of FDG PET/CT in male breast cancers patients (namely patient management). I suggest changing the title in “Diagnostic performance of F18-FDG PET/CT in male breast cancers patients”.
Authors claim that 55.8% of patients were alive. I suggest adding a table indicating survival characteristics of the considered patients.
Minor comments:
Please uniform the terminology, always using F18-FDG PET/CT
Author Response
Re: Manuscript reference diagnostics-1055687
Firstly, I would like to address special thanks and all my gratitude for the efforts and attention accorded to our manuscript.
Please find attached a revised version of our manuscript “Synchronous tumors detected on F18-FDG PET/CT in male breast cancers patients”, which we would like to resubmit for publication as an original article inDiagnostics, special issues "Positron Emission Tomography (PET) Imaging for Therapy Monitoring"
The comments of the reviewers were highly insightful and enabled us to greatly improve the quality of our manuscript. In the following pages are our point-by-point responses to each of the comments of the reviewers.
Revisions in the text are shown using track changes,and strikethrough fontfor deletions. In accordance with reviewer number 1’s suggestions, wehave changed the title. We hope that the revisions in the manuscript and our accompanying responses will be sufficient to make our manuscript suitable for publication.
We shall look forward to hearing from you at your earliest convenience.
REVIEWER: 1 COMMENTS TO THE AUTHOR
“The article is easy to read, and images are very didactical, however English needs corrections.
Thank you distinguished reviewer, we revised the English of the manuscript.
The main drawback of the paper is the title, that is misleading, since the interest of the paper is not only directed to synchronous tumours but covers the role of FDG PET/CT in male breast cancers patients (namely patient management). I suggest changing the title in “Diagnostic performance of F18-FDG PET/CT in male breast cancers patients”.
We changed according to your suggestion.
"Authors claim that 55.8% of patients were alive. I suggest adding a table indicating survival characteristics of the considered patients".
We have synthetized in text the main characteristics of the patients which are alive. We did not introduced a table, due to the fact that the paper mainly focus on the PET/CT contribution.
Minor comments:
"Please uniform the terminology, always using F18-FDG PET/CT"
We have changed.
Reviewer 2 Report
The Abstract could benefit from some editing to streamline its presentation of the study. For example, there is a clear repetition of patient selection. Furthermore, first it is stated that the 23 finally included were obtained from 170 male patients, then that the initial group was of 1113 cases (from which the 23 were then selected). Also, receptor status is reported between these two instances.
It is unclear if the local Institutional Review Board/Ethics Board specifically approved this retrospective study. I am unsure if a generic consent to participate in scientific studies is sufficient without vetting by a Review Board. This point should be clarified to avoid any ethical concerns.
Author Response
Re: Manuscript reference diagnostics-1055687
Firstly, I would like to address special thanks and all my gratitude for the efforts and attention accorded to our manuscript.
Please find attached a revised version of our manuscript “Synchronous tumors detected on F18-FDG PET/CT in male breast cancers patients”, which we would like to resubmit for publication as an original article inDiagnostics, special issues "Positron Emission Tomography (PET) Imaging for Therapy Monitoring"
REVIEWER: 2 COMMENTS TO THE AUTHORS
“The Abstract could benefit from some editing to streamline its presentation of the study. For example, there is a clear repetition of patient selection. Furthermore, first it is stated that the 23 finally included were obtained from 170 male patients, then that the initial group was of 1113 cases (from which the 23 were then selected). Also, receptor status is reported between these two instances.
We have reconsidered the abstract according to your recommendations.
It is unclear if the local Institutional Review Board/Ethics Board specifically approved this retrospective study. I am unsure if a generic consent to participate in scientific studies is sufficient without vetting by a Review Board. This point should be clarified to avoid any ethical concerns.
The institutional regulation do not impose to obtain a ethical approval for the committee in case of retrospective studies, but we have obtained the approval of the study in the Ethics Committee of the site. We have introduced a comment in text.